# A Lightweight Neural Network-Based Method for Detecting Estrus Behavior in Ewes

Longhui Yu [1,2,3,4], Yuhai Pu [1,2,3], Honglei Cen [1,2,3], Jingbin Li [1,2,3,*], Shuangyin Liu [1,4,*], Jing Nie [1,2,3], Jianbing Ge [1,2,3], Linze Lv [1,2,3], Yali Li [1,2,3], Yalei Xu [1,2,3], Jianjun Guo [4], Hangxing Zhao [1,2,3] and Kang Wang [1,2,3]

1 College of Mechanical and Electrical Engineering, Shihezi University, Shihezi 832003, China
2 Xinjiang Production and Construction Corps Key Laboratory of Modern Agricultural Machinery, Shihezi 832003, China
3 Industrial Technology Research Institute of Xinjiang Production and Construction Corps, Shihezi 832000, China
4 College of Information Science and Technology, Zhongkai University of Agriculture and Engineering, Guangzhou 510225, China
* Correspondence: lijingbin@shzu.edu.cn (J.L.); liushuangyin@zhku.edu.cn (S.L.)

**Abstract:** We propose a lightweight neural network-based method to detect the estrus behavior of ewes. Our suggested method is mainly proposed to solve the problem of not being able to detect ewe estrus behavior in a timely and accurate manner in large-scale meat sheep farms. The three main steps of our proposed methodology include constructing the dataset, improving the network structure, and detecting the ewe estrus behavior based on the lightweight network. First, the dataset was constructed by capturing images from videos with estrus crawling behavior, and the data enhancement was performed to improve the generalization ability of the model at first. Second, the original Darknet-53 was replaced with the EfficientNet-B0 for feature extraction in YOLO V3 neural network to make the model lightweight and the deployment easier, thus shortening the detection time. In order to further obtain a higher accuracy of detecting the ewe estrus behavior, we joined the feature layers to the SENet attention module. Finally, the comparative results demonstrated that the proposed method had higher detection accuracy and FPS, as well as a smaller model size than the YOLO V3. The precision of the proposed scheme was 99.44%, recall was 95.54%, F1 value was 97%, AP was 99.78%, FPS was 48.39 f/s, and Model Size was 40.6 MB. This study thus provides an accurate, efficient, and lightweight detection method for the ewe estrus behavior in large-scale mutton sheep breeding.

**Keywords:** behavior detection; deep learning; EfficientNet; ewe estrus; YOLO v3

## 1. Introduction

In recent years, with the increasing quality of living standards and led by healthy diet propaganda, people have gradually been paying more and more attention to green and healthy meat. In meat products, mutton is loved and demanded by people due to its advantages of warm nature, sweet taste, delicate muscle fiber, and richness in protein and amino acids [1,2]. With the increasing demand for mutton, and because the domestic large-scale meat sheep breeding method is still in the primary development stage, making the ewe reproductive capacity level low, the domestic market has a tight supply of mutton and a price surge phenomenon.

Large scale mutton sheep farms, to improve the reproductive ability of ewes is an important measure and guarantee to increase mutton yield and economic benefits. The timely detection of estrus behavior of ewes in large-scale sheep farms is not only a prerequisite for improving the reproductive capacity of ewes, but also a guarantee for ewes to prevent metabolic diseases and individual welfare [3]. At the same time, it can also reduce the cost of feed and insemination in sheep farms.

At present, the traditional manual detection method is mainly used to detect ewe estrus in large-scale mutton sheep farms [4–6]. The manual detection method has the disadvantages of low efficiency, high cost, great subjective influence, and easy false detection. It cannot adapt to the development of precision breeding automation in large-scale mutton sheep farms. Therefore, there is an urgent need for an accurate, efficient, and real-time method to detect the estrous behavior of ewes.

With the rapid development of deep learning technology, researchers have applied deep learning technology to all walks of life [7–12]. In recent years, with the gradual maturity of information technology, precision agriculture, smart agriculture, and precision animal husbandry supported by information technology have been vigorously promoted [13–19]. Researchers have proposed many solutions based on deep learning to solve the problems in the process of agricultural production.

The use of deep learning technology for crop detection [20–26], agricultural management process data forecasting [27–29], crop disease detection and classification [30–39], and animal behavior detection aimed at improving animal welfare has gradually become a research hotspot [40–47]. Wu et al., used the YOLO v3 model to extract the leg region coordinates of the cow in the video frame, calculate the relative step size of the front and rear legs of the cow, construct the relative step size feature vector, and then use the long short term memory networks (LSTM) model to classify the lameness [40]. Yin et al., first used EffcientNet-B0 for feature extraction, then used the weighted bidirectional feature pyramid network (BiFPN) for feature fusion, and finally transmitted the cow behavior feature information to the BiLSTM module to realize the rapid and accurate recognition of individual cow motion behavior [41]. Ho et al., proposed that firstly, the lactation behavior recognition model of sow is constructed with EfficientNet and LSTM, then the R$^3$Det model was used to locate the position of piglets in the video frame, and finally the sort algorithm was used to track the piglets in the video [42]. Wang et al., improved Faster R-CNN to detect goats in surveillance videos by key frame extraction, foreground segmentation and region recommendation. The experimental results show that the accuracy of the improved method was 92.57% [43]. Yang et al., proposed a two-parallel attention module network model in EffcientNet-B2 to analyze the feeding behavior of fish, and used mish activation function, ranger optimizer, and label smoothing training strategies to improve the performance of the model [44]. Duan et al., used long-short term memory (LSTM) networks to classify the sound of short-term feeding behavior of sheep, including biting, chewing, bolus regurgitation, and rumination chewing [45]. According to the characteristics of sheep face data set, Noor et al., used transfer learning method to train CNN model and realized the classification of normal (no pain) and abnormal (pain) sheep images [46]. Song et al., improved the YOLO v3 model by re-clustering anchor frame and channel shearing, and realized the face recognition of sheep [47].

Using deep learning technology to detect animal behavior in real time can not only improve the welfare of animals, but also improve the economic benefits of farmers, as well as promote the development of precision agriculture and precision animal husbandry.

Aiming at the problem that it is difficult to detect the estrous behavior of ewes in time and accurately in large-scale mutton sheep farm. In this paper, an ewe estrus behavior detection method based on lightweight neural network is proposed. Firstly, the ewe oestrus video obtained in the large-scale sheep farm is decomposed into pictures by video key frame decomposition technology. Secondly, LabelImg was used to manually label the pictures to obtain the data set of ewe estrous behavior. Then, aiming at the problem that darknet-53, the feature extraction network of YOLO v3, is slow in feature extraction and accounts for a large proportion of the network scale, a lightweight neural network EfficientNet is proposed as the feature extraction network of YOLO v3. Finally, the three feature layers extracted by the EfficientNet feature extraction network were added to the SENet attention module to further improve the accuracy of the model.

In short, the main contributions of this paper are summarized as follows:

(1) The ewe estrus data set was established, analyzed and processed;

(2) The lightweight neural network EfficientNet was proposed as the feature extraction network of YOLO v3, and the estrous behavior of ewes is detected;

(3) The SENet attention module was added to each of the three feature layers extracted by the EfficientNet feature extraction network to achieve further improvement in the detection accuracy of ewe estrus behavior;

(4) The advantages and disadvantages of the proposed model and other models were compared and verified.

The rest of this paper is organized as follows. In the Section 2, we describe our dataset and propose the important improvement of our network model. In Section 3, we present the experimental results and analysis of the proposed model. Finally, Section 4 concludes the paper and highlights potential future works.

## 2. Materials and Methods

This section describes in detail the materials and methods used in this study, including the collected image data set and the built model.

### 2.1. Data Sources

The video of ewe estrous behavior in this study was collected in Xinao animal husbandry mutton sheep breeding base in Lanzhou Bay Town, Manas County, Xinjiang Uygur Autonomous Region. Through field research, we found that the daily activities of ewes mainly occurred in the shaded area, so this study chose to conduct video collection in the shaded area. The shaded area is 33.75 m in length and 13 m in width, we installed 2 surveillance cameras (Hikvision DS-2CD2346FWD-IS Dome Network Camera) with a resolution of 4 megapixels in the shaded area, with a camera installation height of 2.7 m.

The installed camera and field of view are shown in Figure 1. The video records the activity records of 5 adult rams and 80 ewes, which are collected from May 2021 to November 2021. The video resolution is 2560 pixels × 1440 pixels, and FPS is 25 f/s.

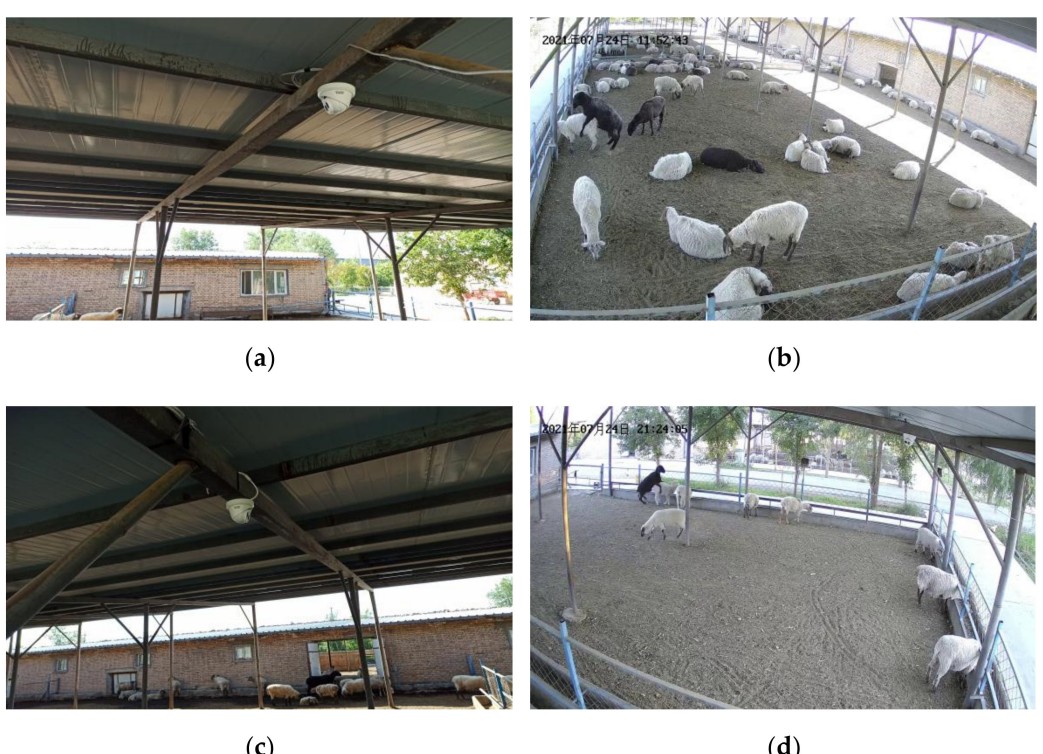

**Figure 1.** Camera installation position and field of view image: (**a**) Camera 1 installation position; (**b**) Camera 1 view image; (**c**) Camera 2 installation position; (**d**) Camera 2 view image. The Chinese in the upper left corner of the image is the date of the video recording, in order of year, month and day.

After the breeding, staff watched the monitoring video of ewe activities, 300 video clips with climbing behavior were manually selected, and the length of each segment ranged from 4 to 20 s. A Python script decomposed the video frame, and one frame was taken every three frames. 3640 images were obtained, in which each image contains not only the straddle behavior of the ewe when it is in estrus, but also other behaviors, such as feeding, standing, walking, lying on the stomach, etc.

To expand the sample size of the dataset and improve the generalization ability of the model, we perform data augmentation operations on the dataset. Firstly, aiming at the uncertainty of the position of the ewe's climbing behavior in the sun shading area during estrus, the data set image was flipped horizontally, moved horizontally, and moved vertically. Secondly, according to the complex and changeable characteristics of sunlight in the shading area, brightness enhancement and brightness reduction are carried out.

After the above data enhancement processing, 5600 data set images were finally obtained. The open-source labeling tool LabelImg was used to label the dataset manually. In this study, we only need to detect the behavior targets such as climbing and spanning behavior. Therefore, the number of labels in this paper is one and the label is named "mounting". The data set is randomly divided according to the ratio of training, verification and test, which is 7:2:1 [20]. There are 3920 images in the training set, 1120 images in the verification set and 560 images in the test set.

### 2.2. Network Model Construction

### 2.2.1. EfficientNet Network Model

During the training of convolutional neural network model, in order to improve the performance of the network model, generally, more layer structure modules were used to increase the depth of the network model [48], increase the number of convolution cores to enlarge the width of the network model [49] and improve the image resolution of the input network [50]. By increasing the depth of the network model, more abundant and complex deep feature information of the target area can be obtained. Increasing the width of the network model can obtain more fine-grained features and it is easier to train. Increasing the resolution of the network model can potentially obtain higher fine-grained feature patterns. These three methods can improve the performance of the network model.

In previous related studies, only one method is usually selected to improve the performance of the network model, which will lead to various problems in the training process of convolutional neural network model. When the network model is too deep, the gradient disappears, the shallow learning ability decreases, and the training difficulty increases. When the model is too wide, it will extract too many repetitive features and reduce deep learning ability. When the model resolution is too large, the accuracy gain will reduce and increase the computational burden of the model. In order to solve the above problems, researchers [51,52] began to explore the relationship between the depth, width and resolution of the network model. Although a large number of experiments have proved that there is a certain relationship between the three dimensions, they have not quantified the substantive mathematical relationship. The EfficientNet network model proposed by Tan et al. [53] in 2019 can effectively solve these problems and mathematically quantify the relationship between the three dimensions. The difference between EfficientNet and traditional methods in network model scaling is shown in Figure 2.

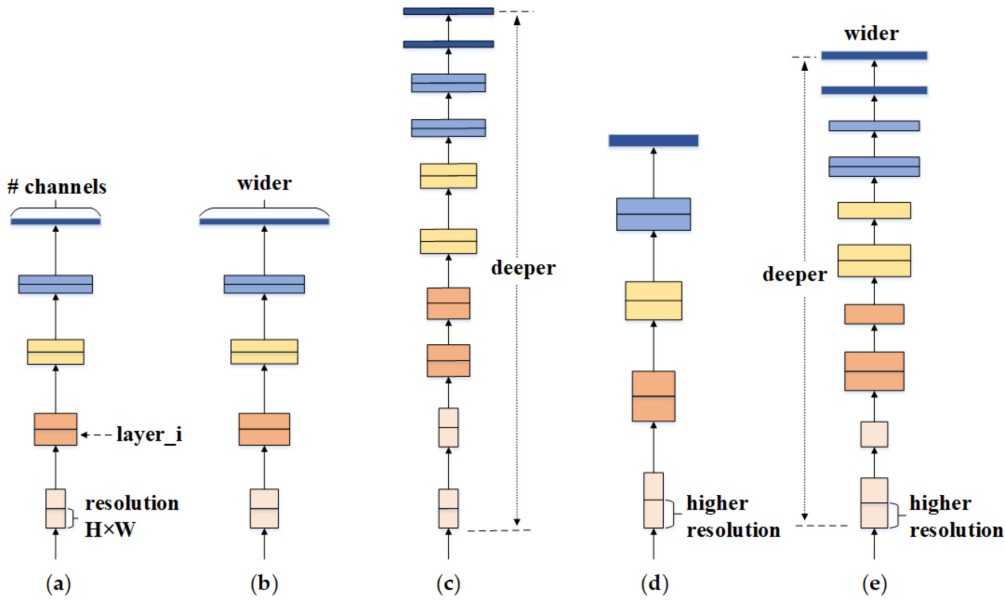

**Figure 2.** Model Scaling: (**a**) baseline network; (**b**) width scaling; (**c**) depth scaling; (**d**) resolution scaling; (**e**) scaling methods for EfficientNet network models.

The baseline network model EfficientNet-B0 of EfficientNet was obtained under certain computing resources by using NAS (Neural Architecture Search) technology [54]. Based on the EfficientNet-B0 network model, a series of EfficientNetB1~B7 network models were obtained by expanding the depth, width and resolution of the model with a constant proportional coefficient see Equations (1) and (2):

$$
\begin{aligned}
\text{depth} : \mathrm{d} &= \alpha^{\phi} \\
\text{width} : \mathrm{w} &= \beta^{\phi} \\
\text{resolution} : \mathrm{r} &= \gamma^{\phi}
\end{aligned}
\tag{1}
$$

$$
\begin{aligned}
\text{s.t. } \alpha \times \beta^2 \times \gamma^2 &\approx 2 \\
\alpha \geq 1, \ \beta \geq 1, \ \gamma &\geq 1
\end{aligned}
\tag{2}
$$

In Equations (1) and (2), d, w, and r are the depth, width and resolution of the network model respectively. Where $\alpha$, $\beta$, and $\gamma$ are constant coefficients of network model depth, width and resolution scaling that can be determined by small grid search. The $\phi$ is a coefficient that can be set; it controls how much computing power and memory can be used to scale the depth, width, and resolution of the model. Generally, the FLOPS of regular convolution operation is directly proportional to d, w and r of the network model, that is, when the depth of the network model was doubled, the FLOPS will be doubled. However, when the width or resolution of the network model was doubled, the FLOPS will be increased by four times. Therefore, we use $\alpha \times \beta^2 \times \gamma^2 \approx 2$ to constrain the FLOPS of the model.

The structure of EfficientNet-B0 is shown in Table 1. Where, Stage is the phase in which the model runs, Operator is the processing mode of each stage, Resolution is the size of feature layer, Stride is the step size of convolution, Channels is the number of feature channels output in each stage, and Layers is the number of layers in each stage.

**Table 1.** EfficientNet-B0 baseline network structure.

| Stage i | Operator $f_i$ | Resolution $H_i \times W_i$ | Stride $S_i$ | # Channels $C_i$ | # Layers $L_i$ |
|---|---|---|---|---|---|
| 1 | Conv3×3 | 224 × 224 | 2 | 32 | 1 |
| 2 | MBConv1, k3×3 | 112 × 112 | 1 | 16 | 1 |
| 3 | MBConv6, k3×3 | 112 × 112 | 2 | 24 | 2 |
| 4 | MBConv6, k5×5 | 56 × 56 | 2 | 40 | 2 |
| 5 | MBConv6, k3×3 | 28 × 28 | 2 | 80 | 3 |
| 6 | MBConv6, k5×5 | 14 × 14 | 1 | 112 | 3 |
| 7 | MBConv6, k5×5 | 14 × 14 | 2 | 192 | 4 |
| 8 | MBConv6, k3×3 | 7 × 7 | 1 | 320 | 1 |
| 9 | Conv1×1 & Pooling & FC | 7 × 7 | 1 | 1280 | 1 |

It can be seen that the whole network model was divided into nine stages. Firstly, the first stage is a normal convolutional layer with a convolutional kernel size of 3 × 3 and a step size of 2; it contains convolution, batch normalization (BN) and Swish activation functions. The second stage to the eighth stage is stacked by using the core MBConv (Mobile Inverted Bottleneck Convolution, MBConv) convolution module. MBConv1 expands the channel dimension of the input characteristic matrix by one time, and MBConv6 expands the channel dimension of the input characteristic matrix by six times. The ninth stage consists of an ordinary convolution layer with a convolution kernel size of 1 × 1 and a step size of 1, a global average pooling layer, and a full connection layer.

The construction of MBConv module was inspired by the deep separable convolution InvertedResidualBlock module in MobileNetV3 model, the difference between the two is that MBConv uses the Swish activation function in the convolution operation and incorporates the SENet (Squeeze-and-Excitation Networks) attention module in the structure. MBConv uses Depthwise Conv and Pointwise Conv operations to extract features, so it has fewer parameters and involves less computation than ordinary convolution. The MBConv structure is shown in Figure 3.

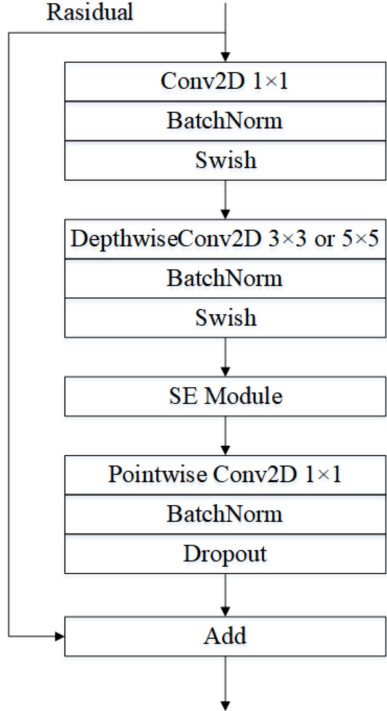

**Figure 3.** MBConv structure diagram.

### 2.2.2. ES_YOLO v3 Model Construction

The YOLO v3 model is a new model proposed by Redmon et al. [55] based on YOLO v2. The feature extraction network for the YOLO v3 model is Darknet-53. The feature extraction of the input image by Darknet-53 is divided into a convolution module and a residual module. The Darknet-53 structure is shown in Table 2.

**Table 2.** Darknet-53 feature extraction network structure.

| Stage $i$ | Operator $f_i$ | Resolution $H_i \times W_i$ | # Channels $C_i$ | # Layers $L_i$ |
|---|---|---|---|---|
| 1 | Conv3×3 | 416 × 416 | 32 | 1 |
| 2 | Conv3×3 | 208 × 208 | 64 | 1 |
| 3 | Conv1×1 Conv3×3 Residual | 208 × 208 208 × 208 | 32 64 | 1 |
| 4 | Conv3×3 | 104 × 104 | 128 | 1 |
| 5 | Conv1×1 Conv3×3 Residual | 104 × 104 104 × 104 | 64 128 | 2 |
| 6 | Conv3×3 | 52 × 52 | 256 | 1 |
| 7 | Conv1×1 Conv3×3 Residual | 52 × 52 52 × 52 | 128 256 | 8 |
| 8 | Conv3×3 | 26 × 26 | 512 | 1 |
| 9 | Conv1×1 Conv3×3 Residual | 26 × 26 26 × 26 | 256 512 | 8 |
| 10 | Conv3×3 | 13 × 13 | 1024 | 1 |
| 11 | Conv1×1 Conv3×3 Residual | 13 × 13 13 × 13 | 512 1024 | 4 |

Although YOLO v3 with darknet-53 as feature extraction network has a very good effect in target detection task, the number of parameters in the Darknet-53 feature extraction network is large and slow in the convolutional extraction operation of the features. The lightweight network model EfficientNet adopts the combination of deep convolution and point-to-point convolution in the convolution extraction of features, and uses the SENet attention module in MBConv. Therefore, EfficientNet network has less computation and fewer parameters than darknet-53 feature extraction network in feature extraction, and one is comparable to another in terms of accuracy. Therefore, this study takes the lightweight network model EfficientNet as the feature extraction network of YOLO v3.

When replacing YOLO v3 feature extraction network Darknet-53 with EfficientNet, three feature layers need to be found for object detection. These three feature layers are the feature layers after Darknet-53 performs the third, fourth, and fifth down-sampling convolution operations on the input image, so the same needs to be done for EfficientNet.

The third down-sampling convolution operation performed by the feature extraction network EfficientNet on the input image is the feature layer of the fourth stage. The fourth down-sampling convolution operation is the fifth stage, but since the convolution step of the sixth stage is 1, and it functions as a feature extraction operation for the feature layer, the feature layer of the sixth stage is used. The fifth down-sampling convolution is the seventh stage, but since the convolution step of the eighth stage is 1, which also performs the feature extraction operation on the feature layer, the feature layer of the eighth stage was used.

The number of channels extracted by the feature extraction network DarkNet-53 is greater than the number of channels extracted by the feature extraction network EfficientNet in the process of extracting three feature layers from the image by the feature extraction network. The feature layer sizes extracted by DarkNet-53 for the input image are (52, 52,

256), (26, 26, 512) and (13, 13, 1024), where 256, 512, and 1024 are the number of channels extracted. If the number of channels extracted by EfficientNet is the same as the number of channels extracted by DarkNet-53, this will increase the number of parameters of the EfficientNet feature extraction network. To further reduce the number of parameters, the number of channels is set according to the number of channels extracted by the original EfficientNet network model 40, 112, and 320.

Finally, the feature extraction network EfficientNet extracts feature layers of dimensions (52, 52, 40), (26, 26, 112), and (13, 13, 320) for the input images, which were used for target detection.

After the replacement of the feature extraction network of YOLO v3 was completed, this paper adds the SENet attention module to the three feature layers extracted by the EfficientNet feature extraction network to improve the network model and further increase the accuracy of the model. The improved network model was named ES_YOLO v3. The SENet structure is shown in Figure 4. The improved network model structure is shown in Figure 5.

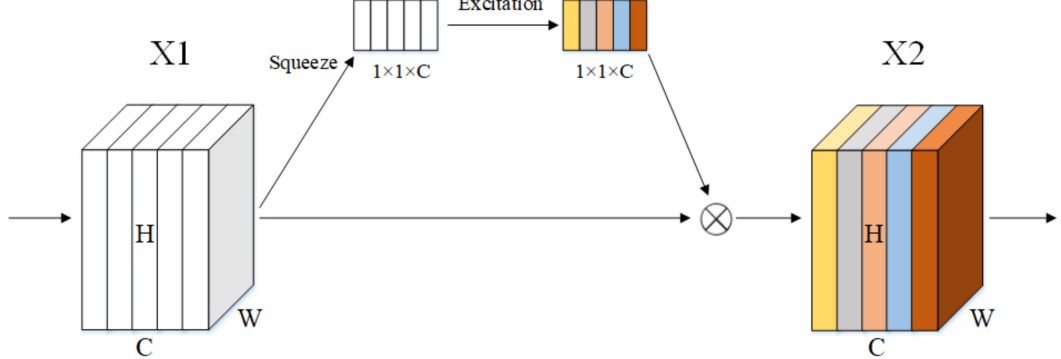

**Figure 4.** SENet structural diagram.

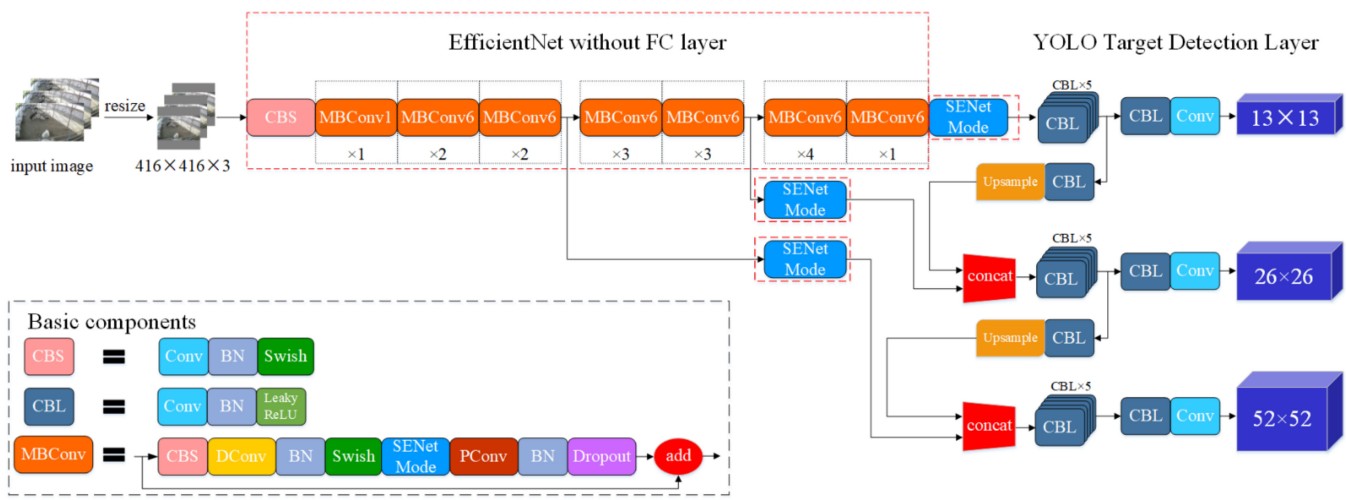

**Figure 5.** ES_YOLO v3 Network Model Structure Diagram.

The feature layer X1 in Figure 4 was obtained by a series of convolutions, and the dimension is H × W × C. Where, H is height, W is width, and C is the number of channels.

The SENet attention module first performs the Squeeze operation on the feature layer X1, mainly to compress H × W × C into 1 × 1 × C, which is equivalent to compressing H × W into one dimension, so that the global feature of the channel will be obtained. This part is usually implemented by global average pooling. Then, perform the Excitation operation, mainly adding the obtained channel global features to a FC fully connected layer.

Learn the relationship between channels, evaluate the importance of each channel, get the weight of each channel, and finally multiply the original feature layer to get a new feature layer X2.

The SENet attention module operates on the channel dimension. This attention mechanism allows the model to pay more attention to the channel features with the most information and suppress the less informative and unimportant channel features, thus increasing the accuracy of model detection.

The ES_YOLO v3 network model first performs a layer of feature extraction and down-sampling convolution operation on the input image through EfficientNet, during which the length and width of the feature layer were compressed and the channel was expanded. Three feature layers will be obtained. Second, these three feature layers were fused with the SENet attention module to form a new feature layer, so that the model can pay more attention to the channel characteristics with the largest amount of information. Then, FPN (Feature Pyramid Networks) operation was performed on three new feature layers. Finally, the feature was connected with the target detection layer to detect the estrus behavior of ewes.

### 2.3. Transfer Learning

Transfer learning makes model learning more stable through knowledge transfer of common features in the convolution layer, thus accelerating model reasoning [56,57]. The initial training weight is the weight of EfficientNet network model trained on ImageNet public data set (The ImageNet dataset is a computer vision dataset that contains 14,197,122 images and 21,841 Synset indexes, each image in the dataset is manually labeled with a category). The initial training weight was transferred to the network model proposed in this study for the initialization of weight parameters. Compared with the random initialization of weights, the use of transfer learning for weight initialization can accelerate the reasoning of the model, reduce the difficulty of training, and further improve the generalization ability of the model [56,57].

### 2.4. Test Evaluation Index

In order to evaluate the performance of the ewe estrus behavior recognition model proposed in this study, the following evaluation indexes were selected: Precision, Recall, Average Precision, mean Average Precision F1, FPS and Model Size, as shown in Equations (3)–(7):

$$P = \frac{T_P}{T_P + F_P} \times 100\% \tag{3}$$

$$R = \frac{T_P}{T_P + F_N} \times 100\% \tag{4}$$

$$F1 = 2 \times \frac{PR}{P + R} \times 100\% \tag{5}$$

$$AP = \int_0^1 (P \times R) dR \tag{6}$$

$$mAP = \frac{\sum_1^n AP}{n} \tag{7}$$

In Equations (3)–(7), $T_P$ represents the number of positive samples identified in the positive samples, $F_P$ represents the number of positive samples identified in negative samples, $F_N$ represents the number of positive samples identified as negative, n represents the number of categories, and P is Precision, R is Recall, F1 is the comprehensive evaluation index of Precision and Recall, AP is Average Precision, and mAP is the mean of Average Precision.

In the evaluation process of the network model, each category can draw the P-R curve according to P and R, and the average precision AP is the area between the P-R curve and

the coordinate axis. The average value of AP of all categories is the value of mAP. Since category n in this paper is 1, AP is also equal to the mAP.

### 3. Results and Discussion

*3.1. Test Platform and Training Parameters Setting*

The model was trained, validated, and tested on the same machine. The computer runs Windows 10 and has a GeForceRTX2080 GPU, an Intel(R) Core (TM) i7-9700k CPU@3.2 GHz, and 32 G of running RAM. Model creation and validation were carried out in Python, utilizing the PyTorch deep learning framework, PyCharm development tools, and a computational framework based on the CUDA 10.0 version (NVIDIA Developer, Santa Clara, CA, USA).

The relevant parameters of the network model during training and validation are set as follows: The batch size of the training sample is 8, the initial learning rate is set to 0.01, the optimizer is SGD, the weight decay is set to 0.0005, the momentum decay is set to 0.9, 100 Epochs are trained, and each Epoch is saved a weight file.

*3.2. Determination of Feature Extraction Network*

The EfficientNet network model includes a baseline model EfficientNet-B0 and an EfficientNet-B1~B7 model obtained by extending the baseline model. We employed the eight models of EfficientNet-B0~B7 as the feature extraction networks of the YOLO v3 model for comparative experiments, to screen out the most suitable feature extraction network.

From Figure 6, we can see that after training 90 Epochs, the loss values of all models only oscillate slightly and gradually stabilize, indicating the completion of the network model training.

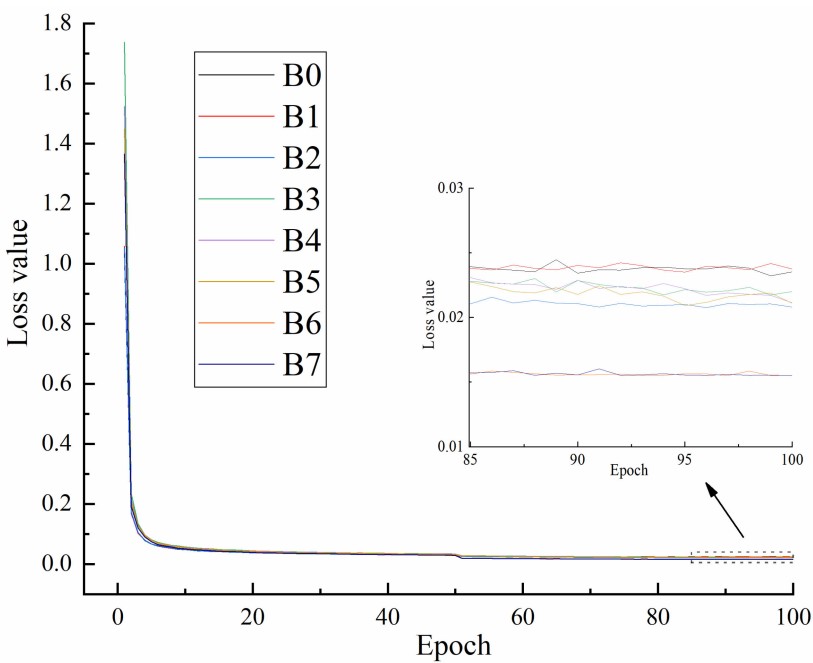

**Figure 6.** The loss value change chart of each model.

From the final loss value curves, we can see that the loss values of the YOLO v3 + Efficientnet-B0~B5 models were not very different from each other and their loss value curves were shaded from each other. The loss values of YOLO v3 + EfficientNet-B6 and YOLO v3 + EfficientNet-B7 models were the smallest due to the fact that the feature extraction net-works EfficientNet-B6 and EfficientNet-B7 have more depth, width, and resolution

than other feature extraction networks. The change of loss values of each model is shown in Figure 6.

After the model training, all the 100 Epoch model weights were selected to test the performance of ewe estrus behavior detection. The test evaluation indexes were F1, AP, P, R, FPS, and Model Size. The indicators related to each model are shown in Table 3.

**Table 3.** Indicators related to each model.

| Feature Extraction Network | F1 | AP | P | R | FPS | Model Size |
|---|---|---|---|---|---|---|
| EfficientNet-B0 | 94% | 98.51% | 98.43% | 89.46% | 50.28 f/s | 40.5 MB |
| EfficientNet-B1 | 94% | 98.49% | 98.81% | 88.75% | 39.76 f/s | 50.2 MB |
| EfficientNet-B2 | 95% | 99.26% | 98.85% | 92.32% | 38.79 f/s | 59.9 MB |
| EfficientNet-B3 | 97% | 99.44% | 98.88% | 94.46% | 37.43 f/s | 77.4 MB |
| EfficientNet-B4 | 97% | 99.40% | 98.52% | 94.82% | 30.05 f/s | 116 MB |
| EfficientNet-B5 | 97% | 99.03% | 99.06% | 94.46% | 26.02 f/s | 172 MB |
| EfficientNet-B6 | 97% | 99.42% | 96.62% | 96.96% | 23.75 f/s | 237 MB |
| EfficientNet-B7 | 97% | 99.37% | 96.60% | 96.94% | 17.61 f/s | 316 MB |

It can be seen from Table 3 that the eight models of EfficientNet-B0~B7 were used as the feature extraction network of YOLO v3 model for the ewe estrus behavior detection test, and good detection results are achieved. Among them, it was found that when the F1 value increased from 94% to 97%, it entered the bottleneck period, and it did not increase after the subsequent model was expanded.

The AP value of the EfficientNet-B3 feature extraction network is the highest, with a value of 99.44%. The P value of the EfficientNet-B5 feature extraction network is the high-est, with a value of 99.06%. The R value of the EfficientNet-B6 feature extraction network is the highest, with a value of 96.96%. The FPS value of the EfficientNet-B0 feature extraction network is the highest, with a value of 50.28 f/s. The value of Model Size mainly increases with the expansion of the feature extraction network, from 40.5 MB of EfficientNet-B0 to 316 MB of EfficientNet-B7. Although the size of the EfficientNet-B7 model is the largest, it is not the best in terms of F1, AP, and P, which may be due to the limitation of the computational power of the computer and the number of GPUs, this is similar to the inference of [39].

Because this study involves the real-time detection of ewe estrus behavior, taking into account the lightweight, speed, and precision of the network model requirements, the FPS, model size, AP, and P of these indicators are the main reference standard.

After determining the main reference indicators, we found that the FPS of Efficient-Net-B0 was far ahead, the model size is also the smallest, and the small model was easier to deploy.

We found that in AP value, the difference between EfficientNet-B0 and the best Effi-cientNet-B3 was 0.93%, and in terms of P value, the difference between EfficientNet-B0 and the best EfficientNet-B5 was 0.63%.

Therefore, after comprehensive consideration, EfficientNet-B0 was used as the feature extraction network of YOLO v3.

### 3.3. ES_YOLO v3 Model Training

In this paper, we add the SENet attention module to the three feature layers extracted by the EfficientNet-B0 feature extraction network, and the improved network model was named ES_YOLO v3, which further improves the performance of the ewe estrus behavior detection model.

Figure 7 shows the variation curve of the loss value of the ES_YOLO v3 model during training. It can be seen that the training loss value of the model in the early training process is rapidly decreasing, which is due to the larger initial learning rate. When the

network model training Epoch takes place more than 89 times, the loss value curve oscillates in a small range, and the overall tends to be gentle, indicating that the model training is completed.

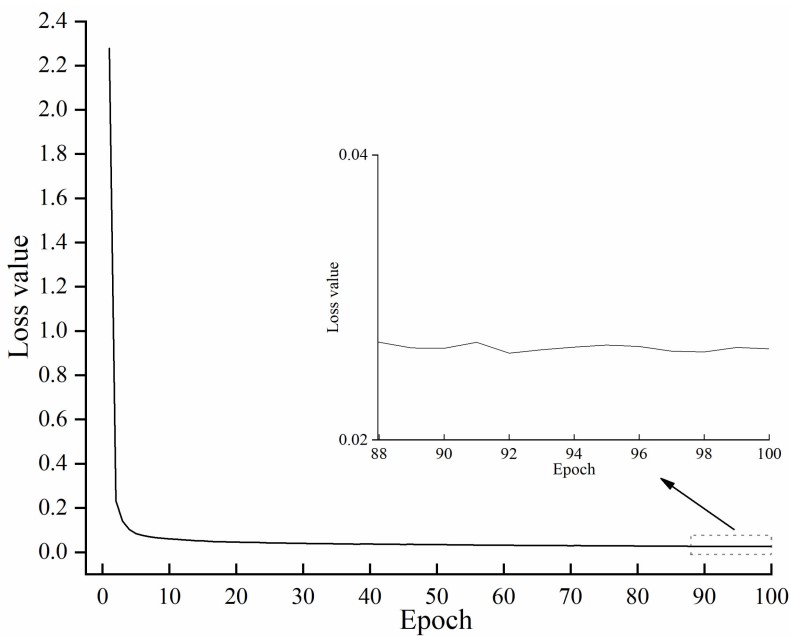

**Figure 7.** Variation of loss value with training times.

The loss value of the ES_YOLO v3 model decreased from an initial value of about 2.3 to about 0.028, indicating the good learning ability of the model for ewe estrus detection, in addition to the slight fluctuation of the final loss value of the model above and below 0.028 indicating the good fitting ability of the model for the study task.

*3.4. Evaluation Index Analysis*

According to the log file, we can draw the change curve of evaluation index with the increase of training Epoch in the training process, as shown in Figure 8.

Figure 8a shows the Precision (P) of the ES_YOLO v3 model for ewe estrus detection, and we can see that the P value of the model quickly reaches a relatively high level in the first 20 Epochs of training. After 40 Epochs, the P value of the model oscillates less and leveled off gradually.

Figure 8b shows the recall (R) of the ES_YOLO v3 model for estrus behavior detection in ewes, and we can see that the R value of the model varies more in the first 89 Epochs of training, but the overall trend is increasing. The R value of the model oscillates less and gradually stabilizes after 89 Epochs.

Figure 8c shows the F1 of the ES_YOLO v3 model for estrus behavior detection in ewes, and we can see that the F1 values of the model are increasing rapidly with small oscillations in the first 89 Epochs of training, and the overall trend is increasing. After 89 Epochs, the F1 values of the model oscillated less and gradually leveled off.

Figure 8d shows the Average Precision (AP) of the ES_YOLO v3 model for estrus behavior detection in ewes. We can see that the AP value of the model rises rapidly in the first 20 Epochs of training and soon reaches a relatively high value. After 60 Epochs, the AP values of the model oscillate less and gradually stabilize.

From Figure 8, we can see that after training 89 Epochs, the ES_YOLO v3 model has less oscillation in each indicator curve and gradually stabilizes.

Taking into account the F1 value and other three indicators, this paper finally selects the model after 100 Epochs as the ewe estrus behavior detection model.

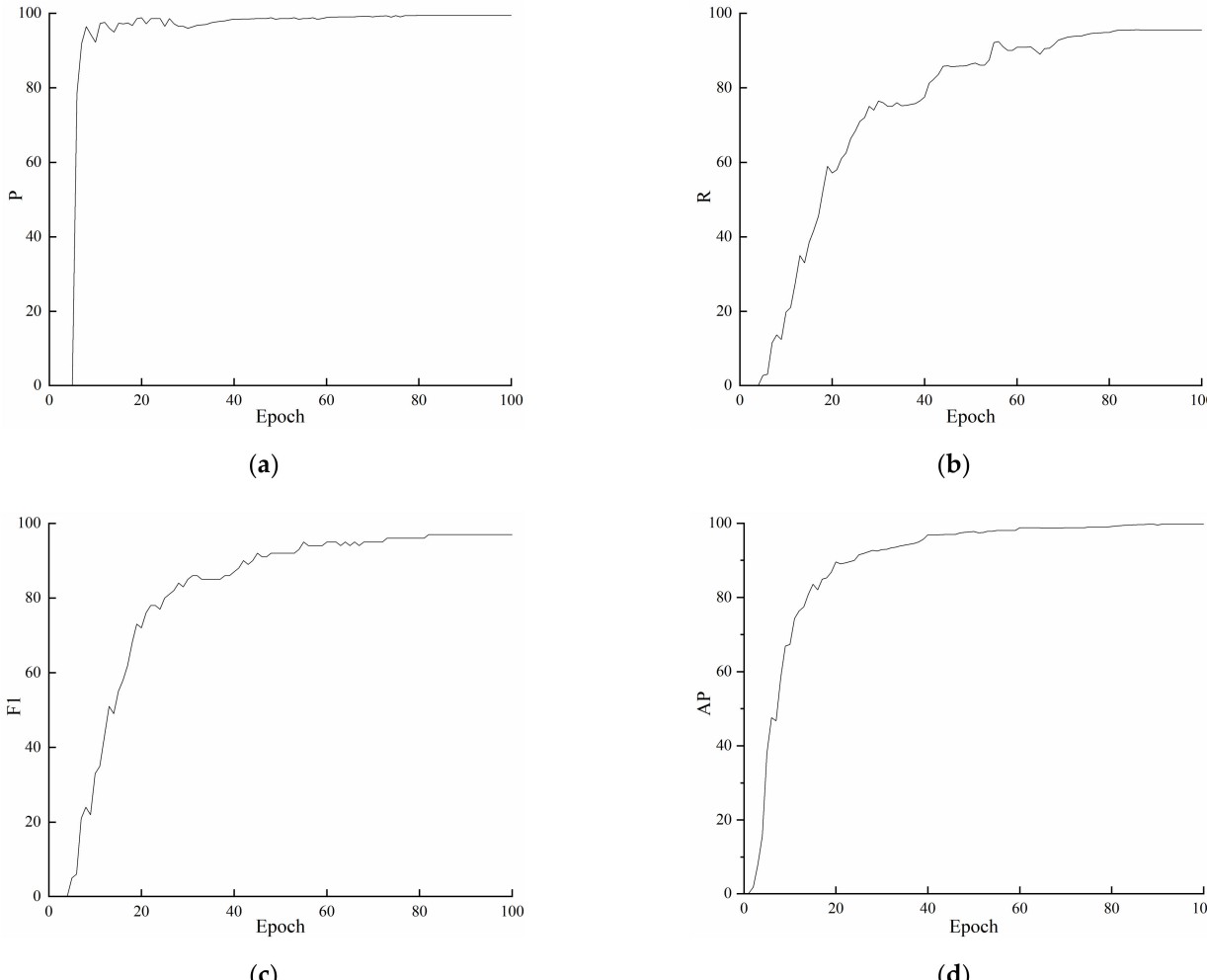

**Figure 8.** Variation of each index with training times: (**a**) Curve of P value changing with training times; (**b**) Curve of R value changing with training times; (**c**) Curve of F1 value changing with training times; (**d**) Curve of AP value changing with training times.

*3.5. Comparative Analysis of Different Confidence Threshold*

After selecting the ewe estrus behavior detection model, we need to compare the F1 value, AP value, P, and R of the model under different confidence thresholds.

It can be seen from Figure 9a that P increases with the increase of confidence threshold, and reaches the maximum when the threshold is 1. It is shown in Figure 9b that R decreases with the increase of confidence threshold and reaches the minimum when the threshold is 1.

When the confidence threshold is set larger, the P of the model is higher and the R is lower, and the estrus detection rate of ewes is higher at this time. When the confidence threshold is set smaller, the R of the model is higher and the missed detection rate is lower, but the precision of estrus detection of ewes is lower. Therefore, the selection of the threshold mainly refers to the comprehensive evaluation index F1 value of precision and recall.

As can be seen from Figure 9c, when the confidence threshold is 0.5, F1 has a higher value, so the confidence threshold of the model is determined to be 0.5, at this time, the P of the model is 99.44%, the R is 95.54%, and the F1 value is 97%. Figure 9d shows that the AP value of the model is 99.78%, when the confidence threshold is 0.5. Since the category in this paper is 1, the mAP value is also 99.78%.

It can be seen that the ES_YOLO v3 model proposed in this paper compared with the YOLO v3 + EfficientNet-B0 model, the F1 of the ES_YOLO v3 model increased by 3%,

the AP increased by 1.27%, the P increased by 1.01%, the R increased by 6.08%, the FPS decreased by 1.89 f/s, and the Model Size increased by 0.1 MB. The above results show the effectiveness and feasibility of added SENet attention module in this paper.

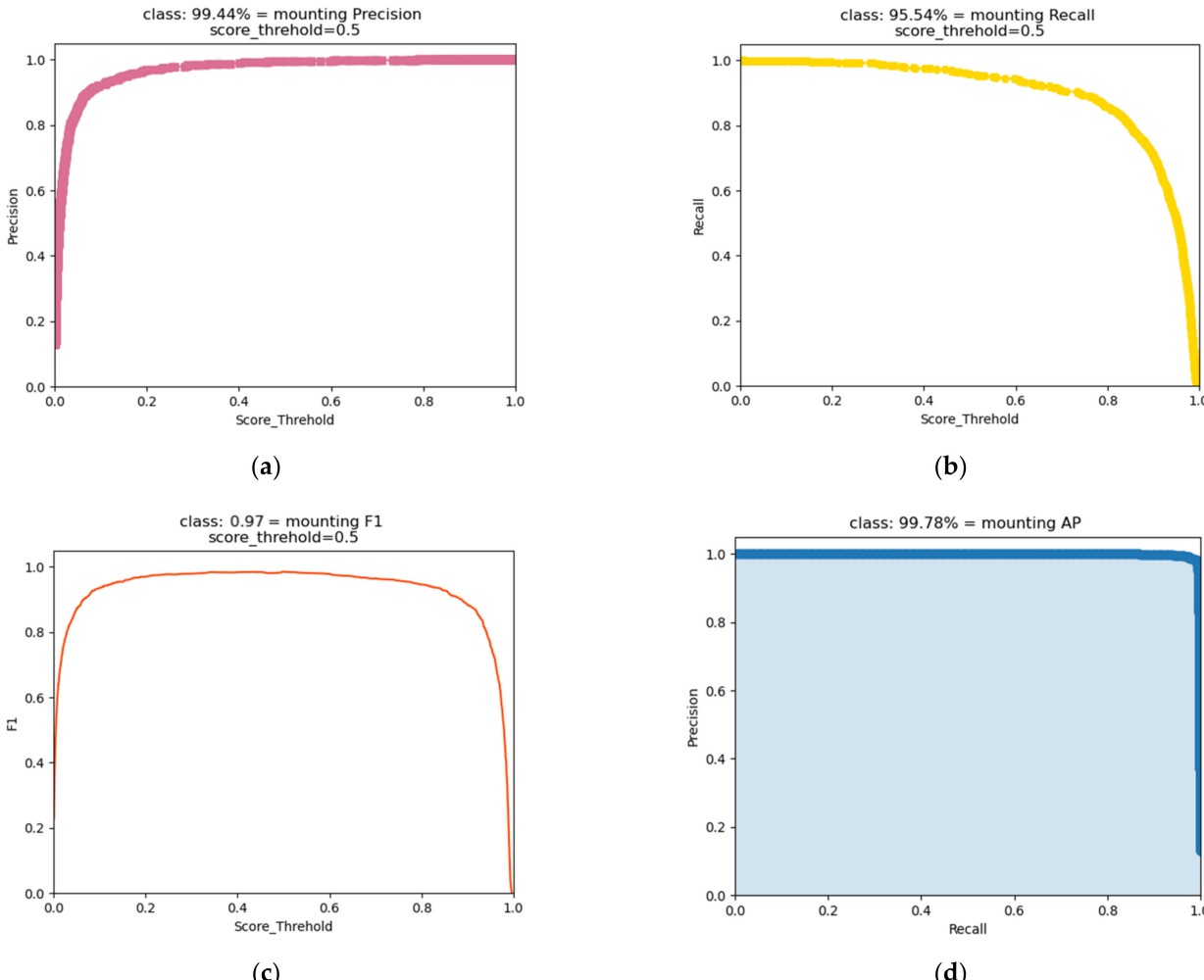

**Figure 9.** Variation of each index with threshold: (**a**) Curve of P value changing with threshold; (**b**) Curve of R value changing with threshold; (**c**) Curve of F1 value changing with threshold; (**d**) Curve of AP value changing with threshold.

*3.6. Comparative Analysis of Different Models*

In order to verify the difference between our proposed ES_YOLO v3 model and other models for estrus behavior detection in ewes, we use Faster R-CNN (Resnet50) [9], Faster R-CNN (VGG16) [58], YOLO v3 [55], and YOLO v4 [59] models to perform comparative experiments on the same test set with P, R, F1, AP, FPS and Model Size evaluation indexes. The results are shown in Table 4.

It can be seen from Table 4, Compared with the Faster R-CNN (Resnet50) model, our proposed ES_YOLO v3 model has a 3% increase in F1, a small difference in AP, a 10.29% increase in P, a 4.28% decrease in R, and a 43.39 f/s increase in FPS, Model Size decreased by 67.4 MB. Compared with Faster R-CNN (VGG16), our model F1 was increased by 8%, AP is not much different, P was increased by 19.24%, R was decreased by 4.28%, FPS was increased by 36.39 f/s, and Model Size was reduced 480.4 MB. Compared with YOLO v3, our proposed model has the same F1, the AP is not much different, the P was increased by 4.58%, the R was decreased by 3.39%, the FPS was increased by 3.39 f/s, and the Model Size was reduced by 194.4 MB. Compared with YOLO V4, our proposed model F1 was

increased by 1%, AP is not much different, P was increased by 1.32%, R was increased by 2.33%, FPS was increased by 12.16 f/s, and Model Size was reduced by 203.4 MB.

**Table 4.** Comparison of recognition effect between this model and other models.

| Evaluation Index | F1 | AP | P | R | FPS | Model Size |
|---|---|---|---|---|---|---|
| Faster R-CNN (Resnet50) | 94% | 99.48% | 89.15% | 99.82% | 5 f/s | 108 MB |
| Faster R-CNN (VGG16) | 89% | 99.41% | 80.20% | 99.82% | 12 f/s | 521 MB |
| YOLO v3 | 97% | 99.29% | 94.86% | 98.93% | 45 f/s | 235 MB |
| YOLO v4 | 96% | 99.14% | 98.12% | 93.21% | 36.23 f/s | 244 MB |
| ES_YOLO v3 | 97% | 99.78% | 99.44% | 95.54% | 48.39 f/s | 40.6 MB |

In summary, our ES_YOLO v3 model has the highest AP and P values in ewe estrus behavior detection, this is because the SENet attention module is added to the three feature layers, which makes the model pay more attention to the channel characteristics with large amount of information and suppress the channel characteristics with small amount of information, so the detection precision of the model is improved. Our model FPS is 48.39 f/s, the detection speed is the fastest, and the Model Size is only 40.6 MB, which is more lightweight than other models. This achievement is because we chose EfficientNet-B0 as the backbone feature extraction network of the ES_YOLO v3 model.

*3.7. Analysis of Identification Results*

We performed the analysis of recognition results in order to further validate the performance of the ES_YOLO v3 model for ewe estrus behavior detection.

Figure 10 shows the effect of near-range detection. The detection of estrus behavior of ewes by the three models is 91%, 99% and 100%. Three models can achieve satisfactory results under the condition of close distance and slight occlusion.

Figure 11 shows the effect of medium distance detection. The three models can detect 81%, 90% and 94% of the behavior of ewes in estrus, respectively. The detection effect of the three models is not ideal, and the reason for this is that two ewes are side by side in the target area to be detected, which interferes with the detection of the models. Despite the related interference, our proposed ES_YOLO v3 model still has the highest detection results.

Figure 12 shows the effect of long-distance detection. The three models can detect 77%, 91% and 96% of the behavior of ewes in estrus respectively. It is shown that the detection results of our proposed ES_YOLO v3 model are still the highest for detection at long distances.

*3.8. Network Attention Visualization*

In order to better understand the ability of our proposed ES_YOLO v3 network model to learn the estrus behavior of ewes, we used Grad-CAM to visualize the estrus behavior detection results of ewes. The visualization effect is shown in Table 5.

It can be seen from the table that the heat maps of YOLO v3 + EfficientNet-B0 and YOLO v3 both highlight the local areas of the estrus behavior of the ewe, but the accuracy of the heat maps is not high and they also contain some irrelevant information. Comparing our proposed ES_YOLO v3 model with YOLO v3 + EfficientNet-B0 and YOLO v3 models, our model can accurately focus on the key regions of ewe estrus behavior, has high accuracy in heat maps, and pays less attention to irrelevant information. Thus, it achieves higher performance than others in ewe estrus behavior detection precision.

Although our proposed model can detect ewe estrus behavior, it does not provide real-time tracking of ewes in estrus and does not allow breeding staff to have specific information on the number and location of ewes in estrus. Our future work is to track ewes in estrus in real time, so that breeding staff can better grasp the number and location of ewes in estrus and other detailed information.

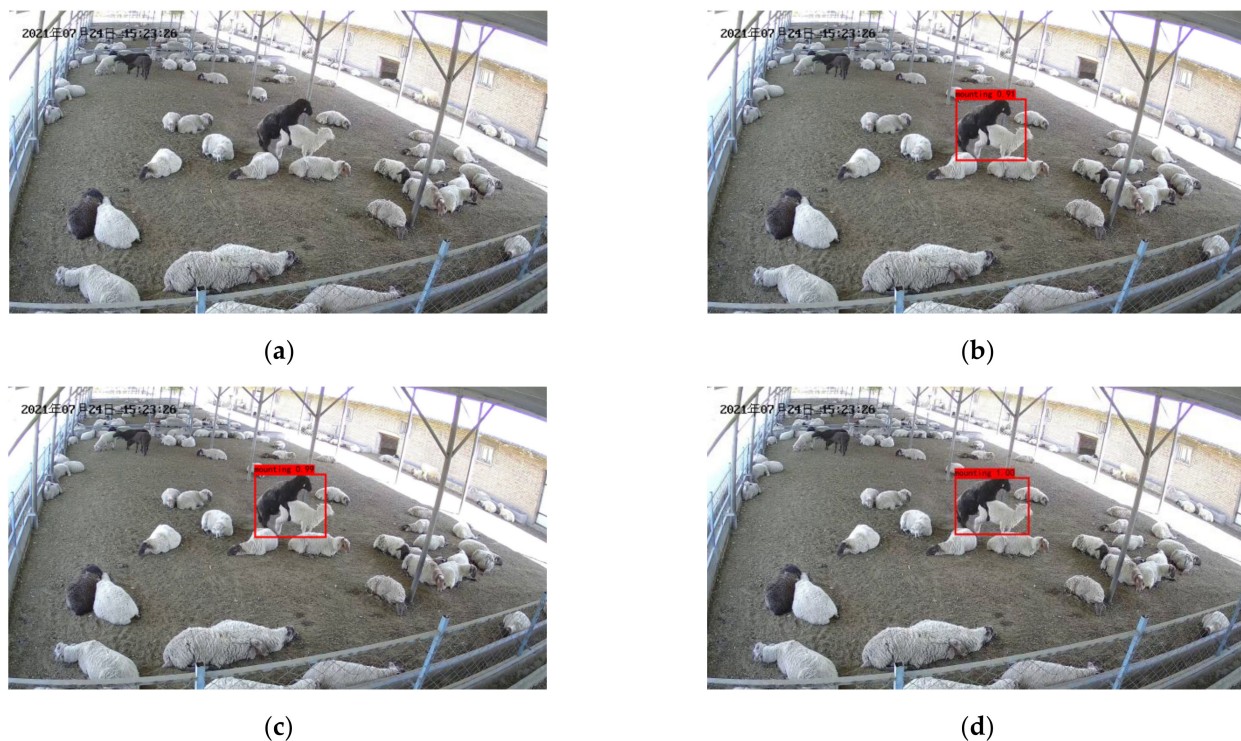

**Figure 10.** Analysis of near-range detection results: (**a**) Original image; (**b**) YOLO v3 + EfficientNet-B0 recognition effect; (**c**) YOLO v3 recognition effect; (**d**) ES_YOLO v3 recognition effect. The Chinese in the upper left corner of the image is the date of the video recording, in order of year, month and day.

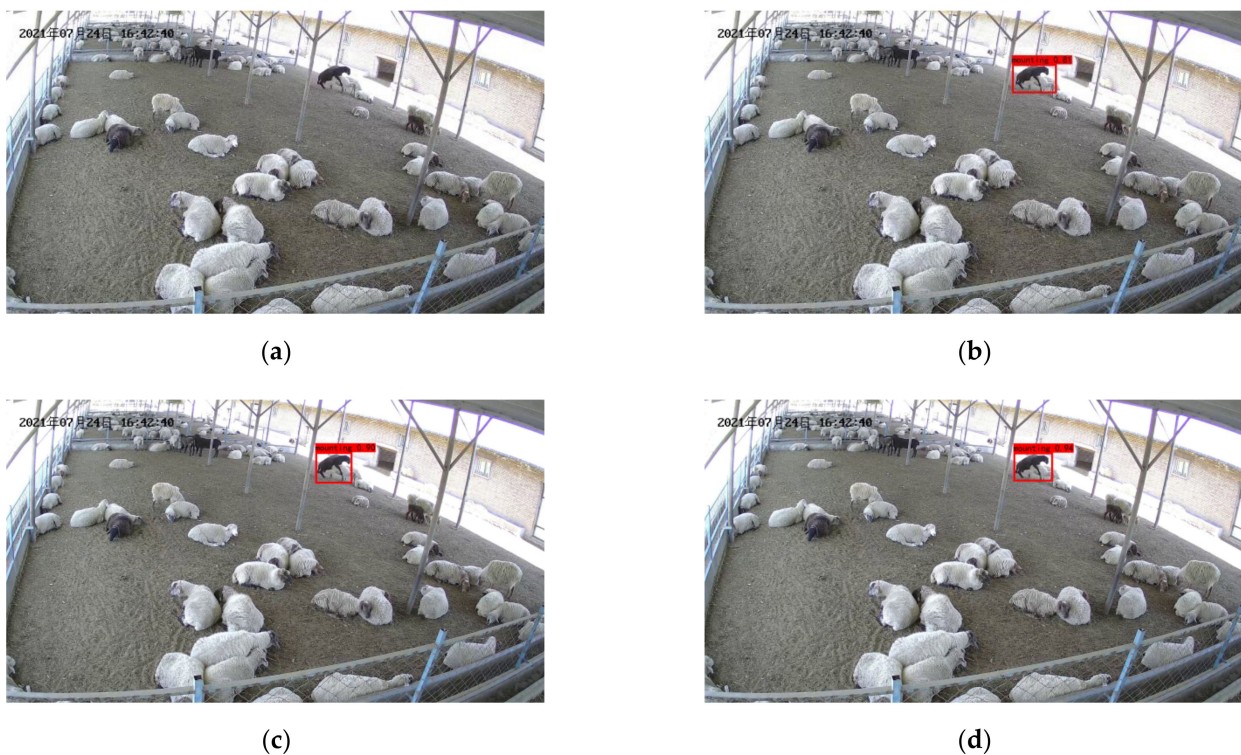

**Figure 11.** Analysis of mid-range detection results: (**a**) Original image; (**b**) YOLO v3 + EfficientNet-B0 recognition effect; (**c**) YOLO v3 recognition effect; (**d**) ES_YOLO v3 recognition effect. The Chinese in the upper left corner of the image is the date of the video recording, in order of year, month and day.

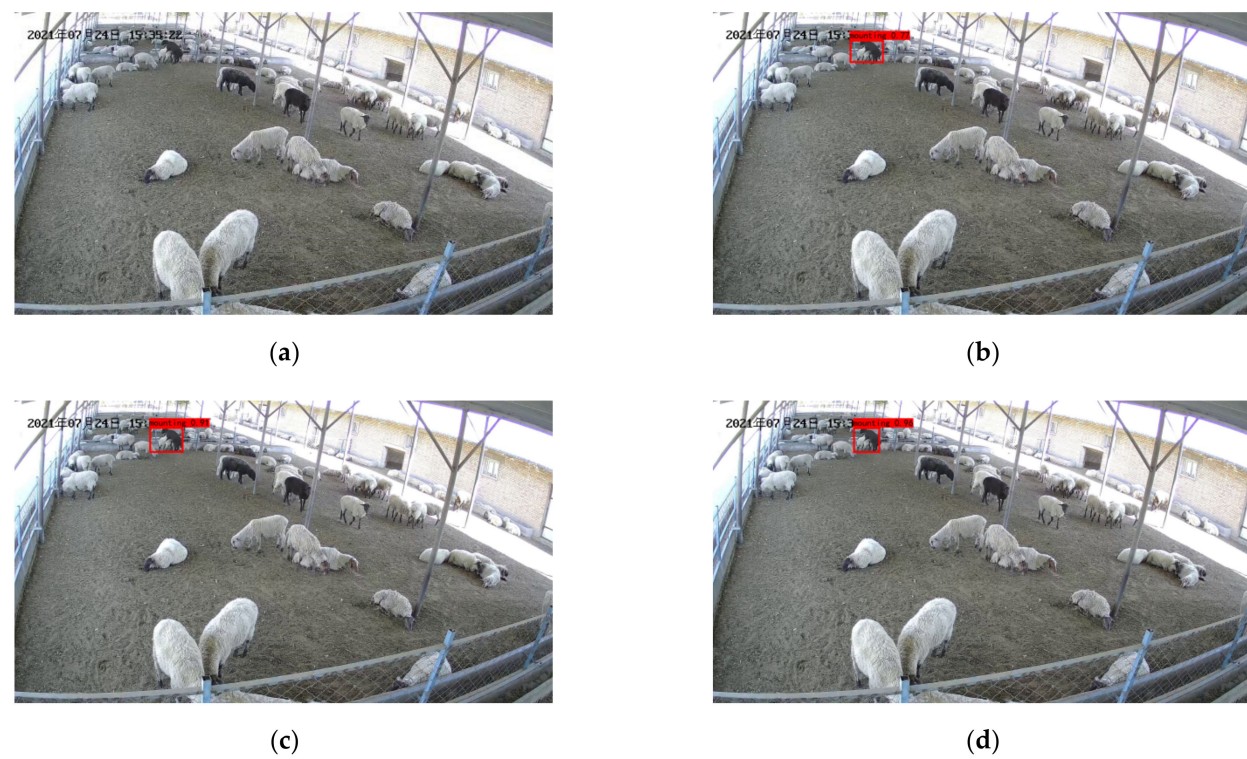

<div align="center">(<b>a</b>)　　　　　　　　　　　　　　　　　　　　　　　(<b>b</b>)</div>

<div align="center">(<b>c</b>)　　　　　　　　　　　　　　　　　　　　　　　(<b>d</b>)</div>

**Figure 12.** Analysis of remote detection results: (**a**) Original image; (**b**) YOLO v3 + EfficientNet-B0 recognition effect; (**c**) YOLO v3 recognition effect; (**d**) ES_YOLO v3 recognition effect. The Chinese in the upper left corner of the image is the date of the video recording, in order of year, month and day.

**Table 5.** Attention heat maps of YOLO v3 + EfficientNet-B0, YOLO v3 and ES_YOLO v3 models in ewe estrus behavior detection. The Chinese in the upper left corner of the image is the date of the video recording, in order of year, month and day.

| YOLO v3 + EfficientNet-B0 | YOLO v3 | ES_YOLO v3 |
|---|---|---|

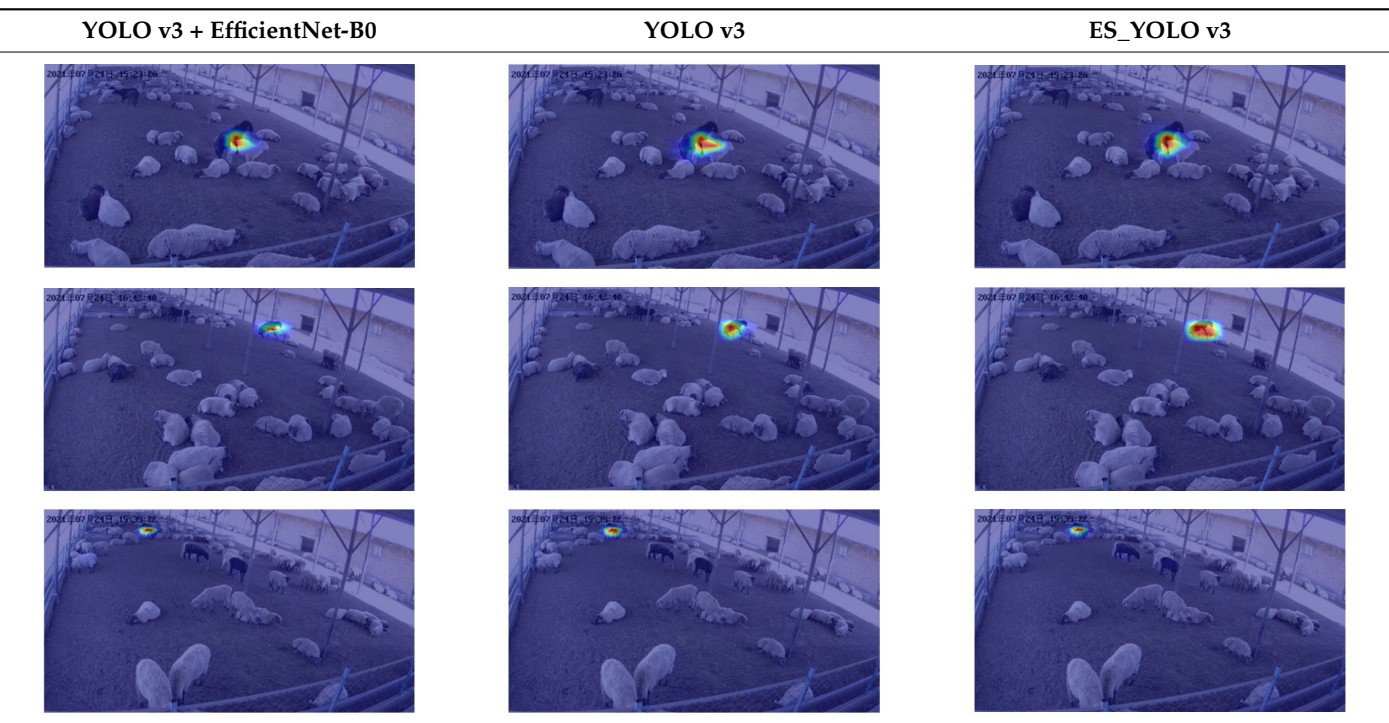

**Table 5.** *Cont*.

| YOLO v3 + EfficientNet-B0 | YOLO v3 | ES_YOLO v3 |
| --- | --- | --- |

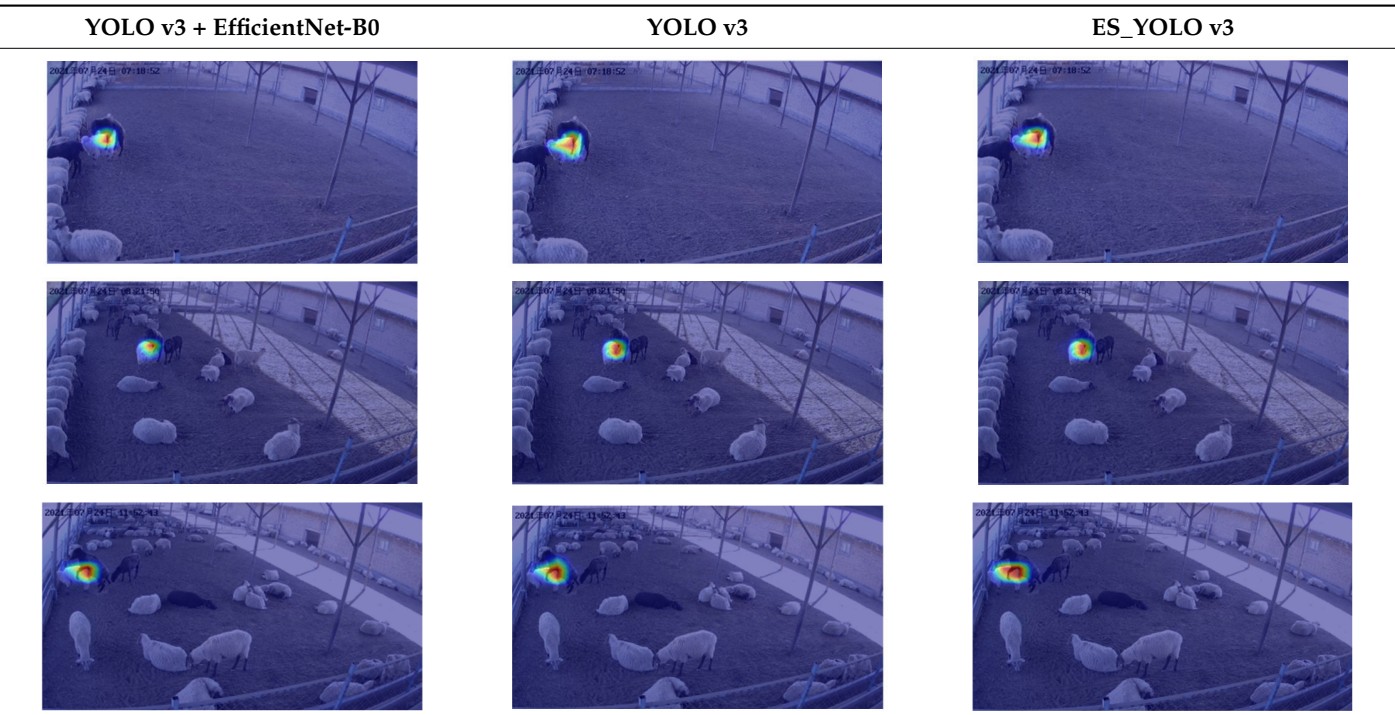

## 4. Conclusions

The timely and accurate detection of estrus behavior of ewes in precision animal husbandry is an important area of research. Our work provides some help for the detection of estrus behavior of ewes in large-scale mutton sheep breeding and proves the following conclusions:

(1) We used EfficientNet-B0~B7 as the feature extraction network of YOLO V3 model and trained the model using transfer learning to obtain the ewe estrus behavior detection model. We found that YOLO V3 + EfficientNet-B0 performed better in the experimental results.

(2) We obtained the ES_YOLO v3 ewe estrus behavior detection model by adding the SENet attention module to the feature extraction network of the YOLO V3 + EfficientNet-B0 model in order to further improve the performance of the model. Our model shows better performance in comparison tests with models, such as Faster R-CNN (Resnet50), Faster R-CNN (VGG16), YOLO v3 and YOLO v4.

In short, we hope that our model can provide some help and reference for estrus behavior detection of ewes in large-scale meat sheep farming.

**Author Contributions:** Conceptualization, L.Y., J.L., and S.L.; data curation, L.Y. and Y.P.; formal analysis, L.Y. and H.C.; funding acquisition, H.C. and J.N.; investigation, L.Y., Y.P., J.G. (Jianbing Ge), L.L., and Y.L.; methodology, L.Y. and Y.X.; project administration, H.C., J.N., and J.G. (Jianjun Guo); resources, L.Y., H.Z., and H.C.; software, L.Y. and K.W.; supervision, J.L. and S.L.; validation, L.Y., J.L., S.L., H.C., J.N., J.G. (Jianbing Ge), and Y.L.; visualization, L.Y. and Y.P.; writing—original draft, L.Y., J.L., and S.L.; writing—review and editing, L.Y., J.L., and S.L. All authors have read and agreed to the published version of the manuscript.

**Funding:** This work is supported by the Shihezi University Achievement Transformation and Technology Promotion Project (Funder: Honglei Cen, Grant Nos. CGZH202103), Shihezi University Innovation and Development Special Project (Funder: Jing Nie, Grant Nos. CXFZ202103), Post expert task of Meat and Sheep System in Agricultural Area of Autonomous Region (Funder: Jie Zhang, Grant Nos. XJNQRY-G-2107), National Natural Science Foundation of China (Funder: Shuangyin Liu, Grant Nos. 61871475), Guangzhou Key Research and Development Project (Funder: Shuangyin Liu, Grant Nos. 202103000033, 201903010043), Innovation Team Project of Universities in Guangdong

Province (Funder: Jianjun Guo, Grant Nos. 2021KCXTD019), Guangdong Province Graduate Education Innovation Program Project (Funder: Jianjun Guo, Grant Nos. 2022XSLT056, 2022JGXM115), Characteristic Innovation Project of Universities in Guangdong Province (Funder: Shuangyin Liu, Grant Nos. KA190578826).

**Institutional Review Board Statement:** Not applicable.

**Informed Consent Statement:** Not applicable.

**Data Availability Statement:** The data presented in this study are available on request from the corresponding author. The data are not publicly available due to these data are part of an ongoing study.

**Acknowledgments:** The authors would like to thank their schools and colleges, as well as the funding of the project. All supports and assistance are sincerely appreciated. Additionally, we sincerely appreciate the work of the editor and the reviewers of the present paper.

**Conflicts of Interest:** The authors declare no conflict of interest.

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
