# Peer review of "A Lightweight Neural Network-Based Method for Detecting Estrus Behavior in Ewes"

_agriculture, doi:10.3390/agriculture12081207_

Round 1

Reviewer 1 Report

The research topic is very interesting. The method can be used successfully to manage the lambing of ewes.

Author Response

Dear Reviewer:

Thanks very much for taking your time to review our manuscript entitled “A lightweight neural network-based method for detecting estrus behavior in ewes” (ID: agriculture-1780870). We are very grateful for your affirmation of our research results and paper, your affirmation has inspired us, and we will continue to work on it. We once again express our heartfelt thanks to you.

Reviewer 2 Report

Dear Authors,

I revised the manuscript "A lightweight neural network-based method for detecting estrus behavior in ewes" submitted to Agriculture journal. The manuscript is interesting with well described methods and results. Unfortunately, the weakest part of the manuscript is the discussion of results. In addition, I have some concerns that need to be addressed.

Minor comments:

1. Section "1. Introduction". I suggest adding more descriptions and references about sheep and machine learning.

2. Line 291. Add more information about the data from the ImageNet dataset.

3. Subsection "2.4 Test Evaluation Index". I suggest adding a confusion matrix.

4. Equations 3 and 4. Each equation must have a separate number.

5. Line 310. I suggest renaming the section from "3. Test Results and Analysis" to "3. Results and Discussion".

6. Line 468. I suggest the section names be corrected to "4. Conclusions".

Major comment:

Section "3. Test Results and Analysis". The discussion of the results should be greatly expanded. You need to compare your results to the results from other papers. Scientific work needs a real discussion of results! Additional references are also needed.

Author Response

Dear Reviewer:

Thanks very much for taking your time to review our manuscript entitled “A lightweight neural network-based method for detecting estrus behavior in ewes” (ID: agriculture-1780870).

We appreciate all your comments and suggestions. Those comments are all valuable and very helpful for revising and improving our paper, as well as the important guiding significance to our research. We have studied the comments carefully and have made the corrections which we hope meet with approval. We have responded to your suggestions as follows:

Point 1: Section "1. Introduction". I suggest adding more descriptions and references about sheep and machine learning.

Response 1: Thank you for your suggestion. We have added references to sheep and machine learning in "1. Introduction" as you suggested and marked them in line 91-98 of the paper.

Point 2: Line 291. Add more information about the data from the ImageNet dataset.

Response 2: Thank you for your suggestion. We have added more information about the ImageNet dataset based on your suggestion and marked it at line 312-314 of the paper.

Point 3: Subsection "2.4 Test Evaluation Index". I suggest adding a confusion matrix.

Response 3: Thank you for your suggestion. We sincerely apologize to you here because we may not have been able to add the confusion matrix in the "2.4 Test Evaluation Index" subsection. The main reason is: We don't understand confusion matrix very well, and there is a big deficiency in the technology of implementing confusion matrix; Since we study the estrus behavior of ewes, there is only one category of "mounting" when making the data set, which leads us to not consider that the confusion matrix can be used to evaluate the model. Here, we once again express our sincere apologies to you.

Point 4: Equations 3 and 4. Each equation must have a separate number.

Response 4: Thank you for your suggestion. We have made changes based on your suggestions and marked the paper 324-325 line.

Point 5: Line 310. I suggest renaming the section from "3. Test Results and Analysis" to "3. Results and Discussion".

Response 5: Thank you for your suggestion. We have made changes based on your suggestions and marked the paper 334 line.

Point 6: Line 468. I suggest the section names be corrected to "4. Conclusions".

Response 6: Thank you for your suggestion. We have made changes based on your suggestions and marked the paper 545 line.

Point 7: Major comment: Section "3. Test Results and Analysis". The discussion of the results should be greatly expanded. You need to compare your results to the results from other papers. Scientific work needs a real discussion of results! Additional references are also needed.

Response 7: Thank you for your suggestion. We have renamed "3. Test Results and Analysis" to "3. Results and Discussion" based on your suggestion. In "3. Results and Discussion", we have made changes according to your suggestions, the main changes are: We have expanded the discussion of the results and marked the 351-359, 418-421, and 425-442 line of the paper; We have added a set of comparison trials and a real discussion of the results as well, marked in the 491-494, and 495 line of the paper.

Reviewer 3 Report

English editing is required by a native editor as there are lots of errors in text.

There is one question about your model: What breed of ewes did you study? How confident are you that your model will work for other breeds, as we are sure that different breeds have different behaviors?

Other comments:

Abstract:

-        Lines 18-20: The starting point of Abstract is too long and confusing, especially this sentence “aiming at the problem that it is hard to detect estrus behavior timely and accurately in …”. Please re-write this sentence.

-        Line 20: “it”? What?

-        Line 21: “extracting” or “detecting” or “predicting”?

-        Line 22: Please delete “Specifically”.

-        Line 26: Please delete “Besides”.

-        Line 27: “respectively”?!

-        Line 29: Please delete “Especially”.

-        Line 30: scheme was …F1 value was ….. was ….. was …..

-        Please be carful with the category of your work: First, second, …. and finally.

Keywords: Your Keywords are not in alphabetic order.

Introduction:

-        Lines 36-37: people …. People

-        Lines 41-42: I fell that something missed here: “and due to the domestic large-scale mutton sheep breeding is still ….”

-        Line 42: “domestic mutton market mutton supply  …” mutton … mutton … mutton … mutton … ??!!

-        Lines 51-52: are you sure that mentioned parameters are “advantages”?!!

-        Line 73: “sows”?

-        Line 74: “ …. was used ….”

-        Line 75: was used …

-        Line 76: To detect what?

-        Line 78: … was 92.57 %.

-        Lines 82-83: but also … but also …!!!

-        Lines 87-93: Firstly ….then …. Secondly…. finally ….Please pay attention to your steps.

-        Line 94: …were added ….

-        Line 98: … was proposed …

-        Line 103: …were compared …

Materials and Methods:

-        Line 122: …. And the …?

-        Line 125: After the breeding, staff …

-        Line 128: 3640 images were obtained …

-        Line 134: data set image was flipped …

-        Line 135: . Secondly,

-        Line 137: 5600 data set images were finally …

-        Line 142: Why you use 7:2:1 ratio for training, verification and test? Reference?

-        Lines 144-207: Are you sure that this section is required for a “research article”??

-        Line 210: Please don’t start your sentence with “And”.

-        Lines 208-284: Section 2.2.2. is too long to introduce construction of ES_YOLO v3. Please present a brief of this model as it is your “Materials and Methods”.

-        Lines 286-287: Please delete this sentence.

-        Lines 293-295: Reference?

-        Line 299: …and Model Size, as shown …

-        Line 300: There are 5 equations, but you just mentioned 3 and 4?

Results and Discussion

-        Line 314: Model creation and validation were carried out …

-        Line 341: …network is the highest

-        Lines 341-342: Please correct this sentence: The model with ….., with a value of 99.06%;

-        Line 342: What is “;”?!!! Please finish your sentence with “.” not “;”. You repeated it in other places of the text (Lines 345, 357, …).

-        Line 347: Please delete “It is not difficult to find that

-        Lines 351-357: It is a big paragraph, difficult to follow-up with readers. Please correct it.

-        Line 358: … best EfficientNet-B5 was 0.63%.

-        Line 359: Many tines you used “is” instead of “was”.

Conclusions:

-        Lines 474-478: Please delete this paragraph as repeated in other parts of your manuscript.

-        Lines 479 and 486: What is (1) and (2)?

-        Please summarize the “Conclusions” section with the main points of your findings.

Author Response

Dear Reviewer:

Thanks very much for taking your time to review our manuscript entitled “A lightweight neural network-based method for detecting estrus behavior in ewes” (ID: agriculture-1780870).

We appreciate all your comments and suggestions. Those comments are all valuable and very helpful for revising and improving our paper, as well as the important guiding significance to our research. We have studied the comments carefully and have made the corrections which we hope meet with approval. We have responded to your suggestions as follows:

Point 1: English editing is required by a native editor as there are lots of errors in text.

Response 1: Thank you for your suggestion. We have corrected the errors in the text based on your suggestion and asked a native speaker to check the essay. We would like to thank you very much for your help with the paper.

Point 2: There is one question about your model: What breed of ewes did you study? How confident are you that your model will work for other breeds, as we are sure that different breeds have different behaviors?

Response 2: We thank you very much for this question. We give you a brief answer here: We mainly studied the detection of estrus behavior of Suffolk sheep, Dorper sheep and hybrid sheep. In our study, we found some different behaviors in these ewes during estrus, but we additionally found that all of these breeds of ewes show the behavior of mounting across during estrus, so we have more confidence that the proposed model can be used to detect estrus behavior in other breeds of sheep.

We are very grateful to you for this question, which has led to great inspiration for our future work. We intend to study the estrus behavior of different breeds of ewes in more detail in future work, hoping to further improve the generalization ability of our model. 

Abstract:

Point 3: Lines 18-20: The starting point of Abstract is too long and confusing, especially this sentence "aiming at the problem that it is hard to detect estrus behavior timely and accurately in …". Please re-write this sentence.

Response 3: Thank you very much for your suggestion. We have made changes based on your suggestion and marked the paper 20-22 line.

Point 4: Line 20: "it"? What?.

Response 4: Thank you very much for your suggestion. The "it" refers to the method we propose. We have made changes based on your suggestion and marked the paper 23 line.

Point 5: Line 21: "extracting" or "detecting" or "predicting"?.

Response 5: Thank you very much for your suggestion. We have revised "extracting" to "detecting" based on your suggestion, and marked in the paper 24 line.

Point 6: Line 22: Please delete "Specifically".

Response 6: Thank you very much for your suggestion. We deleted "Specifically" as you suggested and added “First” here, and marked in the paper 25 line.

Point 7: Line 26: Please delete "Besides".

Response 7: Thank you very much for your suggestion. We have removed "Besides" as per your suggestion, and marked in the paper 29 line.

Point 8: Line 27: "respectively"?!.

Response 8: Thank you very much for your suggestion. We have removed "respectively" as you suggested, and marked in the paper 31 line.

Point 9: Line 29: Please delete "Especially".

Response 9: Thank you very much for your suggestion. We have removed "Especially" as you suggested, and you can see it in the paper in the 33 line.

Point 10: Line 30: scheme was …F1 value was … was … was …..

Response 10: Thank you very much for your suggestion. We have changed the word "is" to "was" according to your suggestion, and marked in the paper 33-34 line.

Point 11: Please be carful with the category of your work: First, second, …. and finally.

Response 11: Thank you very much for your suggestion. We have made careful changes according to your suggestions, where "First" can be seen in line 25, "Second" in line 27, and "Finally" in line 31 of the paper.

Point 12: Keywords: Your Keywords are not in alphabetic order.

Response 12: Thank you very much for your suggestion. We have sorted the keywords in alphabetical order according to your suggestions, and marked in the paper 37-38 line.

Introduction:

Point 13: Lines 36-37: people …. People.

Response 13: Thank you very much for your suggestion. We have made changes based on your suggestions and marked in the paper 41-43 line.

Point 14: Lines 41-42: I fell that something missed here: "and due to the domestic large-scale mutton sheep breeding is still ….".

Response 14: We are very grateful for your suggestion. We have made an addition based on your suggestion, the addition is "making the ewe reproductive capacity level is low" and marked in the paper 49 line.

Point 15: Line 42: "domestic mutton market mutton supply  …" mutton … mutton … mutton … mutton … ??!!.

Response 15: We are very grateful for your suggestion. We have made changes based on your suggestions, marked in the paper 47-50 line.

Point 16: Lines 51-52: are you sure that mentioned parameters are "advantages"?!!.

Response 16: We are very grateful for your suggestion. We have changed the word "advantages" to "disadvantages" according to your suggestion, and marked in the paper 61 line.

Point 17: Line 73: "sows"?.

Response 17: We are very grateful for your suggestion. We have changed the word "sows" to "sow" according to your suggestion, and marked in the paper 83 line.

Point 18: Line 74: " … was used … ".

Response 18: We are very grateful for your suggestion. We changed "is used" to "was used" based on your suggestion, and marked in the paper 84 line.

Point 19: Line 75: was used …

Response 19: We are very grateful for your suggestion. We changed "is used" to "was used" based on your suggestion, and marked in the paper 85 line.

Point 20: Line 76: To detect what?

Response 20: We are very grateful for your suggestion. We will make a supplement according to your suggestions. The supplementary content is "detect goats in surveillance videos", and marked in the paper 86 line.

Point 21: Line 78: … was 92.57 %.

Response 21: We are very grateful for your suggestion. We changed "is" to "was" based on your suggestion, and marked in the paper 88 line.

Point 22: Lines 82-83: but also … but also …!!!

Response 22: We are very grateful for your suggestion. We changed "but also" to "as well as" based on your suggestion, and marked in the paper 100-101 line.

Point 23: Lines 87-93: Firstly ….then …. Secondly…. finally ….Please pay attention to your steps.

Response 23: Thank you very much for your suggestion. We have made careful changes according to your suggestions, where "Firstly" can be seen in line 105, "Secondly" in line 107, "Then" in line 108, and "Finally" in line 111 of the paper.

Point 24: Line 94: … were added … .

Response 24: Thank you very much for your suggestion. We have changed the word "are" to "were" according to your suggestion, and marked in the paper 112 line.

Point 25: Line 98: … was proposed ….

Response 25: We are very grateful for your suggestion. We have changed the word "is" to "was" according to your suggestion, and marked in the paper 116 line.

Point 26: Line 103: …were compared …. 

Response 26: We are very grateful for your suggestion. We have changed the word "are" to "were" according to your suggestion, and marked in the paper 122 line.

Materials and Methods:

Point 27: Line 122: …. And the …?

Response 27: We are very grateful for your suggestion. We have revised according to your suggestion and marked the 140 line of the paper.

Point 28: Line 125: After the breeding, staff ….

Response 28: We are very grateful for your suggestion. We have added " , " as per your suggestion, and marked in the paper 144 line.

Point 29: Line 128: 3640 images were obtained ….

Response 29: We are very grateful for your suggestion. We have changed the word "are" to "were" according to your suggestion, and marked in the paper 147 line.

Point 30: Line 134: data set image was flipped ….

Response 30: We are very grateful for your suggestion. We have changed the word "is" to "was" according to your suggestion, and marked in the paper 153 line.

Point 31: Line 135: . Secondly.

Response 31: We are very grateful for your suggestion. We changed ";" to "." based on your suggestion, and marked in the paper 154 line.

Point 32: Line 137: 5600 data set images were finally ….

Response 32: We are very grateful for your suggestion. We have changed the word "are" to "were" according to your suggestion, and marked in the paper 156 line.

Point 33: Line 142: Why you use 7:2:1 ratio for training, verification and test? Reference?

Response 33: We are very grateful for your suggestion. We introduce reference [20] based on your suggestion, and marked in the paper 161 line.

Point 34: Lines 144-207: Are you sure that this section is required for a "research article"??

Response 34: We thank you very much for this question. We answer here, Our proposed model is inspired by the EfficientNet Network Model, so we feel that this part cannot be ignored and describe it in the article.

Point 35: Line 210: Please don’t start your sentence with "And".

Response 35: We are very grateful for your suggestion. We have revised according to your suggestion and marked the 229-231 line of the paper.

Point 36: Lines 208-284: Section 2.2.2. is too long to introduce construction of ES_YOLO v3. Please present a brief of this model as it is your "Materials and Methods".

Response 36: We are very grateful for your suggestion. We have made some deletion based on your suggestion, and marked in the 233-238 line of the paper.

We did not delete many paragraphs in the "2.2.2. ES_YOLO v3 Model Construction" section. We hereby express our sincere apologies to you. The main reason is: We feel that if the paragraphs in the article are deleted too much, it may make the article more concise, and it may also make the article reading experience not good. We once again express our sincere apologies to you.

Point 37: Lines 286-287: Please delete this sentence.

Response 37: We are very grateful for your suggestion. We have removed this sentence based on your suggestion, and marked in the 309-311 line of the paper.

Point 38: Lines 293-295: Reference?

Response 38: We are very grateful for your suggestion. We introduce reference [56-57] based on your suggestion, and marked in the paper 318-319 line.

Point 39: Line 299: …and Model Size, as shown ….

Response 39: We are very grateful for your suggestion. We changed " . " to " , " based on your suggestion, and marked in the paper 323 line.

Point 40: Line 300: There are 5 equations, but you just mentioned 3 and 4?

Response 40: We are very grateful for your suggestion. We have revised according to your suggestion, and marked in the 324-325 line of the paper.

Results and Discussion:

Point 41: Line 314: Model creation and validation were carried out ….

Response 41: We are very grateful for your suggestion. We changed " are " to " were " based on your suggestion, and marked in the 338 line of the paper.

Point 42: Line 341: …network is the highest.

Response 42: We are very grateful for your suggestion. We have revised according to your suggestion, and marked in the 373-380 line of the paper.

Point 43: Lines 341-342: Please correct this sentence: The model with ….., with a value of 99.06%.

Response 43: We are very grateful for your suggestion. We have revised according to your suggestion, and marked in the 373-380 line of the paper.

Point 44: Line 342: What is " ; "?!!! Please finish your sentence with " . " not " ; ". You repeated it in other places of the text (Lines 345, 357, …).

Response 44: We are very grateful for your suggestion. We changed " ; " to " . " based on your suggestion, and marked in the 373-380 and 397-399 line of the paper.

Point 45: Line 347: Please delete "It is not difficult to find that".

Response 45: We are very grateful for your suggestion. We have removed it based on your suggestion, and marked in the 387 line of the paper.

Point 46: Lines 351-357: It is a big paragraph, difficult to follow-up with readers. Please correct it.

Response 46: We are very grateful for your suggestion. We have revised according to your suggestion, and marked in the 393-404 line of the paper.

Point 47: Line 358: … best EfficientNet-B5 was 0.63%.

Response 47: We are very grateful for your suggestion. We changed " is " to  " was " based on your suggestion, and marked in the 398-399 line of the paper.

Point 48: Line 359: Many tines you used "is" instead of "was".

Response 48: We are very grateful for your suggestion. We checked the article in detail according to your suggestions.

Conclusions:

Point 49: Lines 474-478: Please delete this paragraph as repeated in other parts of your manuscript.

Response 49: We are very grateful for your suggestion. We have removed it based on your suggestion, and marked in the 546-550 line of the paper.

Point 50: Lines 479 and 486: What is (1) and (2)?

Response 50: We are very grateful for your suggestion. We rewrite this section according to your suggestion, and marked in the 572-580 line of the paper.

Point 51: Please summarize the "Conclusions" section with the main points of your findings.

Response 51: We are very grateful for your suggestion. We have rewritten the "Conclusions" section according to your suggestion, and marked in the 568-582 line of the paper.

We once again express our heartfelt thanks to you.

Round 2

Reviewer 3 Report

All comments have been addressed.